# LRRK2 Inhibition Mitigates the Neuroinflammation Caused by TLR2-Specific α-Synuclein and Alleviates Neuroinflammation-Derived Dopaminergic Neuronal Loss

**DOI:** 10.3390/cells11050861

**Published:** 2022-03-02

**Authors:** Dong-Hwan Ho, Daleum Nam, Mikyoung Seo, Sung-Woo Park, Wongi Seol, Ilhong Son

**Affiliations:** 1InAm Neuroscience Research Center, Wonkwang University, Sanbon-ro 321, Gunpo-si 15865, Gyeonggi-do, Korea; ekfma304@naver.com (D.N.); wseolha@gmail.com (W.S.); 2Paik Institute for Clinical Research, Inje University College of Medicine, Busan-si 47392, Korea; first1011486@hanmail.net (M.S.); neuro109@hanmail.net (S.-W.P.); 3Department of Convergence Biomedical Science, Inje University College of Medicine, Busan-si 47392, Korea; 4Department of Neurology, Sanbon Medical Center, College of Medicine, Wonkwang University, Sanbon-ro 321, Gunpo-si 15865, Gyeonggi-do, Korea

**Keywords:** Parkinson’s disease, α-synuclein, leucine-rich repeat kinase 2 (LRRK2), neuroinflammation

## Abstract

Evidence suggests that crosstalk occurs between microglial leucine-rich repeat kinase 2 (LRRK2)—a regulator of neuroinflammation—and neuron-released α-synuclein (αSyn)—a promoter of microglial activation and neuroinflammatory responses—in neuroinflammation-mediated Parkinson’s disease (PD) progression. Therefore, we examined whether LRRK2 inhibition reduces the responses of microglia to neuroinflammation caused by neuron-released αSyn. We examined the neuroinflammatory responses provoked by Toll-like receptor 2 (TLR2)-positive αSyn of neuronal cells using an LRRK2 inhibitor in the mouse glioma cells, rat primary microglia, and human microglia cell line; and the effects of LRRK2 inhibitor in the co-culture of ectopic αSyn-expressing human neuroblastoma cells and human microglia cells and in mouse models by injecting αSyn. We analyzed the association between LRRK2 activity and αSyn oligomer and TLR2 levels in the substantia nigra tissues of human patients with idiopathic PD (iPD). The TLR2-specific αSyn elevated LRRK2 activity and neuroinflammation, and the LRRK2 inhibitor ameliorated neuroinflammatory responses in various microglia cells, alleviated neuronal degeneration along with neuroinflammation in the co-culture, and blocked the further progression of locomotor failure and dopaminergic neuronal degeneration caused by TLR2-specific αSyn in mice. Furthermore, LRRK2 phosphorylation was increased in patients with iPD showing αSyn-specific high TLR2 level. These results suggest the application of LRRK2 inhibitors as a novel therapeutic approach against αSyn-mediated PD progression.

## 1. Introduction

Neuroinflammatory responses play a crucial role in the defense mechanisms of the brain [1]. However, neuroinflammation may act as a double-edged blade because the release of excessive quantities of pro-inflammatory cytokines worsens the degeneration of neurons in patients with neurodegenerative diseases, which accelerates the disease progression [2]. The number of reactive microglia is higher in the brains of patients with Parkinson’s disease (PD) [3], and pro-inflammatory cytokine levels are higher in the biofluids, such as the cerebrospinal fluid and plasma, of the former than those of the latter [4,5]. The stimulator of neuroinflammation in patients with PD remains elusive, but previous studies have demonstrated that the release of α-synuclein (αSyn) from neurons is responsible for the activation of microglia via the Toll-like receptors (TLRs), CD36, or P2X7 receptor [6,7,8]. Furthermore, αSyn is known as a major component of Lewy bodies, and oligomeric αSyn is involved in both dopaminergic neuronal degeneration and neuroinflammatory responses [9].

Leucine-rich repeat kinase 2 (LRRK2) is considered a modulator of neuroinflammation because its kinase activity regulates microglial reactivity and the levels of pro-inflammatory cytokines [10,11]. In particular, the stimulation of TLRs using specific ligands was shown to induce LRRK2 activity in microglia in a previous study [12]. Moreover, our previous study showed that the upregulated kinase activity of G2019S LRRK2 mutants promoted microglia-derived neuroinflammation [13].

Previous studies revealed that a neuron-derived αSyn oligomer triggered a TLR2-specific activation of microglia and their neuroinflammatory response [14,15]. Because the stimulation of TLR2 results in the upregulation of LRRK2 activity in microglia, we speculated that TLR2-specific αSyn released from dopaminergic neurons enhanced the neuroinflammatory responses via LRRK2 activity in microglia. Furthermore, the inhibition of LRRK2 activity could alleviate the degeneration of dopaminergic neurons by mitigating excessive neuroinflammatory responses in microglia. In this study, we demonstrated that LRRK2 inhibition is important in the treatment of αSyn-mediated neuroinflammation in patients with PD.

## 2. Materials and Methods

### 2.1. Cell Culture and Treatment

The mouse microglial cell line, BV2, was cultured in Dulbecco’s modified Eagle medium (DMEM; 10-013-CV, Corning, Cellgro, Thermo Fisher Scientific, Waltham, MA, USA), 5% fetal bovine serum (FBS; BFS-1000 500 mL, T&I, Seoul, Korea), and 1% penicillin–streptomycin (P/S; 10378016, Gibco, Thermo Fisher Scientific) with 0.1 μg/mL mycoplasma removal agent (MRA, KOMA biotechnology, 093050044). The human microglial cell line, C20, was cultured in DMEM/F12 (LM 002–08, Welgene, Daegu, Korea), 10% FBS, 1% P/S, and 0.1 μg/mL MRA. The human neuroblastoma cell line, SH-SY5Y, was cultured in DMEM, 10% FBS, 1% P/S with 0.1 μg/mL MRA, and 10 μM all-trans retinoic acid in DMSO (RA, R2625, Sigma, St. Louis, MO, USA) for 7 days to produce differentiated SH-SY5Y (dSH), which turned into dopaminergic neuron-like cells. dSH cells were transfected on day 6 with vector plasmids or Flag-tagged αSyn for 24 h; we replaced the culture media with DMEM alone and incubated the cells for 24 h. The media were centrifuged at 4000× *g* for 10 min at 4 °C; we then concentrated the media using a 10 K membrane filter unit to make 10× conditioned media of vector (Vt-CM) and αSyn (αS-CM). BV2 or C20 cells were treated with 1 μg/mL Pam3CSK4 (P3C4, tlrl-pms, InvivoGen, San Diego, CA, USA), 0.2 μg/mL isolated αSyn proteins, and αS-CM containing 0.2 μg/mL αSyn or an identical volume of Vt-CM for 18 h. To verify the effects of LRRK2 inhibition, we treated cells with 1 μM of the LRRK2 inhibitor, GSK2578215A (4629, TOCRIS, Bristol, UK), or MLi-2 (TOCRIS, 5756).

### 2.2. Generation and Isolation of Oligomeric αSyn

We bought endotoxin-removed recombinant human αSyn (AS-55555-1000, Anaspec, Fremont, CA, USA) and incubated 2 mg/mL for 5 days at 4 °C. The heterogenous αSyn mixture was subjected to fast protein liquid chromatography (FPLC, Amersham Pharma Biotech, Piscataway, NJ, USA) using Superose^®^ 6 10/300 GL (GE29-0915-96, GE, Piscataway, NJ, USA), after which we assessed the αSyn concentration in fractions following size-exclusion chromatography using αSyn enzyme-linked immunosorbent assay (ELISA) as detailed in our previous report [16]. We used fraction 8 as an oligomeric αSyn and fraction 13 as a monomeric αSyn.

### 2.3. Western Blot and Dot-Blot Analyses

We lysed cells with 1× sample buffer (50 mM Tris-HCl, pH 6.8, 2% sodium dodecyl sulfate, 10% glycerol, 1% β-mercaptoethanol, and 0.02% bromophenol blue). After sodium dodecyl sulphate–polyacrylamide gel electrophoresis (SDS-PAGE), proteins in SDS-gel were transferred to nitrocellulose membranes. Denatured samples from cell lysates or undenatured samples from the isolated fraction of recombinant αSyn were subjected to a nitrocellulose membrane. Details of the Western blot and dot-blot analyses were previously described [17]. Visualization and densitometry of protein bands were performed using mouse anti-αSyn (#42, 610786, BD biosciences, San Jose, CA, USA, 1:1000), mouse anti-αSyn oligomer-specific (clone number 2.4, ASyO5, AS132718, Agrisera, Vännäs, Västerbotten, Sweden, 1:1000), rabbit anti-pS935 LRRK2 (ab133450, Abcam, Cambridge, UK, 1:1000) mouse anti-LRRK2 (N241A/34, 75–253, NeuroMab, Davis, CA, USA, 1:400), mouse anti-Rab10 (Abcam, ab104859, 1:500), rabbit anti-phospho-T75 Rab10 (Abcam, ab230261, 1:1000), rabbit anti-pS1292 LRRK2 (Abcam, ab203181. 1:200), mouse anti-α-tubulin (DM1A, T5168, Sigma, 1:20,000) mouse anti-CD68 (3F103, sc-70761, Santa cruz, Dallas, TX, USA, 1:500), rabbit anti-TLR2 (LS-B1417, LSBio, Seattle, WA, USA, 1:500), and rabbit anti-αSyn filament conformation-specific (ab209538, MJFR-14-6-4-2, Abcam, 1:1000) antibodies, peroxidase-conjugated AffiniPure Goat Anti-Mouse IgG (H + L) (115-035-003, Jackson Immunoresearch Laboratories Inc., West Grove, PA, USA, 1:5000), and peroxidase-conjugated AffiniPure Goat Anti-Rabbit IgG (H + L) (111-035-144, Jackson Immunoresearch Laboratories Inc. 1:5000). Membranes for Western blotting and dot blotting were developed using Immobilon Crescendo Western HRP substrate (WBLUR0500, Merck, Kenil Worth, NJ, USA) and images were captured using MicroChemi (DNR).

### 2.4. Human TLR2 Activity Test

We bought and incubated the human TLR2-expressing HEK-293 cells, HEK-Blue™hTLR2 (hkb-htlr2, InvivoGen), as per the manufacturer’s instructions. After using the same conditions of BV2 treatment for 4 h, we measured nuclear factor kappa B (NF-κB)-mediated SEAP activities using QUANTI-Blue™ (rep-qbs, InvivoGen) by reading absorbance at 655 nm. We regarded treatment with P3C4 as a control for 100% hTLR2 activity and examined the results of other treatments. Synergy™ 2 (Biotek, Winooski, VT, USA) was used to measure absorbance.

### 2.5. Isolation and Culture of Rat Primary Microglia

For culturing, rat primary microglial cells were obtained from fetal brains {embryonic day 17 (E17)}; dissection of the brains and isolation of cells were performed as described in our previous study [18].

### 2.6. Enzyme-Linked Immune Sorbent Assay (ELISA) of Pro-Inflammatory Cytokine

To assess the levels of cytokines released in microglial cells of the three species, we used the following commercial cytokine ELISA kits: mouse tumor necrosis factor α (mTNFα, 430905, BioLegend, San Diego, DA, USA), mouse IL-1β (mIL-1β, BioLegend, 432605), rat TNFα (DY510-05, R&D Systems, Minneapolis, MN, USA), and human TNFα (Abcam, ab181421). We also used the Griess Assay kit (G7921, Invitrogen, Thermo Fisher Scientific) to detect extracellular nitrite levels. The experiments were conducted using the protocols provided with the kits, and the released levels were assessed according to each standard curve.

### 2.7. Immunofluorescence

Cells were fixed with 4% paraformaldehyde and permeabilized with 0.1% Triton X-100 in PBS. Proteins were visualized using mouse anti-CD68 (1:50), rabbit anti-pS935 LRRK2 (3H8L19, 701066, Invitrogen, 1:300), rabbit anti-β-tubulin III (T2020, Sigma, 1:50), and mouse anti-β-actin (sc-47778, Santa Cruz, 1:100) antibodies. Hoechst 33342 was used to stain the nuclei of cells. The fluorescence intensities of proteins were evaluated after normalizing them to Hoechst 33342 intensity. Neurite lengths were estimated using the NeuronJ plugin of ImageJ. Images were captured with a confocal microscope LSM 700 (Ziess, Oberkochen, Germany).

### 2.8. Co-Culture of C20 with dSH

Differentiation of SH-SY5Y cells was performed with 10 μM RA with DMEM/F12, 10% FBS, and 1% P/S for 7 days. The transfection of vector and Flag-tagged αSyn plasmids was performed on differentiation day 4. On the day after transfection, dSH cells were cultured with fresh culture media. Transfected dSH cells were detached and seeded on C20 cells on day 6 of SH-SY5Y differentiation and stabilized for 24 h with the culture media without RA. On the subsequent day, we applied the DMEM/F12 only condition to the co-culture cells for 24 h. Cells were fixed with 4% paraformaldehyde and permeabilized with 0.1% Triton X-100 in PBS. Anti-β-tubulin III antibody, anti-β-actin antibody, and Hoechst 33342 were then used for the staining of dSH neurites, total cell bodies, and nuclei, respectively. The cultured media were subjected to LDH activity assay to assure cytotoxicity or kits for human TNFα and nitrite release.

### 2.9. Mouse Handling and Post-Injection Behavioral Testing

All care processes were conducted in accordance with the provisions of the Dankook Animal Ethics Committee (Dankook IACUC, 18-026). A 15-week-old male C57BL/6J (Daehan Biolink, Eumseong, Korea) was placed in a new cage and acclimatized for one week. A solution obtained by mixing ketamine, xylazine, and physiological saline in a ratio of 1:1:8 was injected intramuscularly in mice at a dose of 1.5 mL/kg. Vt- or αS-CM were injected into the substantia nigra (SN) pars compacta located at AP: −3 mm, ML: 1.2 mm, and DV: −4.5 mm from the bregma using a CMA12 microdialysis probe (8309662, CMA Microdialysis AB, Kista, Sweden). A total dose of 1 μL was injected at the rate of 0.25 µL/min for 4 min. After performing the injection, we waited for 5 min without removing the probe to allow sufficient absorption of Vt- or αS-CM. Then, MLi-2 was dissolved in PEG400 and sterilized in distilled water at a 1:1 ratio; 2 mg/kg of MLi-2 was administered intraperitoneally to mice microinjected with αS-CM. The other groups were intraperitoneally administered with DMSO in a volume equivalent to that of MLi-2 in the same solvent. MLi-2 and DMSO were administered once every two days and continued for two weeks to minimize the intraperitoneal injection stresses.

Each mouse was placed in the center of a square (30 cm × 30 cm) space for the open-field test. The experiment was recorded for 30 min using a camcorder. When the recording was finished, ANY-MAZE (Stoelting) was used to track the movement of the mouse, and the total distance and mean speed (m/s) of each movement were calculated.

### 2.10. Immunohistochemistry (IHC) and Immunocytochemistry (ICC)

The tissues were cut in the coronal direction at a thickness of 20 μm using a microtome (SM3050S, Leica, Wetzlar, Germany). For immunohistochemistry, tissues were washed with 0.1 M PBS and incubated at 48 °C with rabbit anti-tyrosine hydroxylase (Abcam, ab112, 1:500), mouse anti-alpha-synuclein oligomer-specific (ASyO5, 1:500), anti-pS935 LRRK2 (LS-B1417, Thermo Fisher, 1:50), and anti-TLR2 antibodies (1:40). The sections were incubated with secondary antibodies for 1 h in an avidin-biotin-peroxidase solution (PK-6100, Vector Laboratories, Burlingame, CA, USA) at a ratio of 1:250 and then stained with 3′-diaminobenzidine. For immunocytochemistry, tissues were blocked by adding 10% bovine serum albumin to PBS containing 0.2% Triton X-100. To stain the microglia, the sections were incubated with anti-rabbit Iba1 (010-19741, Wako, Fujifilm Wako Pure Chemical Corporation, Osaka, Japan) antibodies at 4 °C for a day at 1:400 with goat anti-rabbit IgG (1:200) at room temperature for 1 h and then stained with 4′6-diamidino-2-phenylindole (1:2000). The stained tissue was observed under a microscope (BX 51, Olympus, Tokyo, Japan). The optical density (OD) of tissue sections was estimated using ImageJ version 1.52a software (NIH).

### 2.11. Human Substantia Nigra Tissue Extraction

Donated SN tissues from control patients or patients with idiopathic PD (iPD) were obtained from the SNUH Brain Bank (Seoul, Korea), and their clinical summary can be found in our previous report [19]. We lysed SN tissues with M-PER™ Mammalian Protein Extraction Reagent (Thermo, 78503) and homogenized the samples using Kontes™ Pellet Pestle Motor (Sigma, Z359971). Debris were removed by centrifugation at 2000 rpm for 10 min at 4 °C, and the supernatants were collected for ELISA analysis and mixed with 5× sample buffer for the dot-blot or Western blot assay.

### 2.12. Data Analysis and Presentation

Densitometry was performed using a Multi Gauge version 3.0 software (Fujifilm, Fujifilm Wako Pure Chemical Corporation), and statistical analyses and production of graphs were performed using Prism 8 (GraphPad, La Jolla, CA, USA). Every data set was analyzed with a one-way analysis of variance and Bonferroni’s or Tukey’s post hoc test, and asterisks are used to indicate the following: * *p <* 0.05, ** *p <* 0.01, *** *p <* 0.001, and **** *p <* 0.0001. All data are represented as the mean ± SEM.

## 3. Results

### 3.1. Upregulation of LRRK2 Activity Is Promoted by TLR2 Activation via the αSyn Oligomer

Previous studies revealed that the treatment of BV2 cells with P3C4, the ligand for TLR2, increased LRRK2 activity [12], and the TLR2-specific αSyn enhanced microglial activation and neuroinflammation in a previous study [14,15]. Hence, we assumed that the TLR2-specific αSyn elevated microglial activity via the upregulation of LRRK2 activity. According to a previous report [14], we generated and purified TLR2-specific αSyn from endotoxin-removed recombinant αSyn. We collected an oligomer fraction (number 8 fraction indicated with the red letter in Figure 1A). 

Western blot analysis showed that oligomeric αSyn was mainly composed of the high molecular weight (HMW) αSyn, and a dot-blot analysis via the antibody-recognizing oligomeric αSyn revealed sensitive signals (Figure 1B,C). This fraction of oligomeric αSyn led to a significant increase in human TLR2 (hTLR2) activity (Figure 1D). Before performing the experiments using the collected αSyn oligomer, we tested whether the LRRK2 inhibitor could alleviate the phosphorylation of serine 935 in LRRK2, which represents the upregulation of LRRK2 activity by P3C4 in BV2 cells. We found that LRRK2 inhibitor (GSK-KI) decreased P3C4-mediated LRRK2 activation (Appendix A) and the release of mTNF (Appendix A), but not that of mIL-1β (Appendix A). Regarding monomeric and oligomeric αSyn treatment of BV2 cells with or without GSK-KI, we observed that only oligomers and not monomers were responsible for the increase in LRRK2 activity, and co-treatment with an LRRK2 inhibitor reduced LRRK2 activity (Figure 1E,E). We confirmed that the changes in mTNF-release following treatment with oligomeric αSyn or co-treatment with LRRK2 inhibitor were similar to those caused by LRRK2 activity (Figure 1G). However, the increase in mIL-1β-release by the oligomeric αSyn was not reduced by LRRK2 inhibition (Figure 1H). Therefore, these results collectively indicate that the TLR2-specific αSyn oligomer might be critical for LRRK2 activation-mediating neuroinflammation and that an LRRK2 inhibitor could mitigate the neuroinflammation stimulated by TLR2-specific αSyn.

### 3.2. Activation of TLR2 by Neuron-Released αSyn Is Responsible for the Increase in LRRK2 Activity

To reliably support the aforementioned results using the intentionally synthesized TLR2-specific αSyn oligomer, the following experiments were conducted using the αSyn derived from dSH cells similar to those used in a previous study [14]. We collected and concentrated the αS-CM or Vt-CM. For the analysis of αS-CM, αS-CM was estimated using our established sandwich ELISA method for total αSyn, and 200 ng of αSyn in αS-CM was used in the following experiments. We validated the presence of the HMW Flag-αSyn oligomer in αS-CM using Western blotting (Figure 2A) and dot-blotting (Oligomer AyO5 antibody in Figure 2B).

Moreover, hTLR2 activity was significantly induced by αS-CM treatment (Figure 2C). Treatment of αS-CM promoted an elevation of LRRK2 activity in BV2 cells, and co-treatment with LRRK2 inhibitor decreased LRRK2 activity (Figure 2D,E). With regard to αS-CM treatment, only a significant decrease in mTNFα level was found following co-treatment with LRRK2 kinase inhibitor (Figure 2F), and there was no difference in mIL-1β release (Figure 2G). However, we could not disregard the effect of the conditioned media derived from dSH, because the Vt-CM also showed about 50% of the activity of hTLR2 (Figure 2C). Hence, we performed immunoprecipitation of the Flag tag (Flag-IP) on αS-CM. We confirmed that most of Flag-αSyn was cleared in the immunoprecipitated-conditioned media (IPed) and captured in the Flag-IP fraction (Appendix A). The level of Flag-αSyn was significantly lower in IPed αS-CM by approximately 94.8% than in Pre-IP (Appendix A), and IPed αS-CM exhibited a 64.7% decrease in human TLR2 activity (Appendix A). Treatment of BV2 cells with IPed αS-CM significantly decreased LRRK2 activity and mTNFα level (Appendix A). To verify our findings, we carried out similar experiments using rat primary microglia. Treatment with αS-CM accelerated LRRK2 phosphorylation (pS935 LRRK2) (Figure 3A,B,E,F) and increased neuroinflammatory marker levels, including rat-released TNFα levels (Figure 3G), CD68 expression (Figure 3A,C), and microglial activation leading to ameboid morphology (Figure 3A,D). 

Phosphorylation levels of the LRRK2 substrate Rab10 (pT75 Rab10), which serves as a surrogate of LRRK2 kinase activity, were also elevated by αS-CM and by the treatment of GSK-KI (Figure 3E,F). We validated the identical effects of LRRK2 inhibitor on TLR2 agonist (P3C4)-mediated LRRK2 activation and microglial activation (Appendix A). The microglial TLR2-targeting neuroinflammation and LRRK2 kinase activations were significantly alleviated by co-treatment with an LRRK2 kinase inhibitor (Figure 3, Appendix A). We also observed that TLR2 stimulation mediated by P3C4 or αS-CM in human microglial C20 cells increased the autophosphorylation of serine 1292 in LRRK2 (pS1292) and the CD68 levels, and LRRK2 inhibitor reduced these changes (Figure 4A–F). 

However, we could not find any significant differences, except between the αS-CM with the inhibitor group and αS-CM without inhibitor group. The increased release levels of human TNFα and nitrite following the stimulation of C20 using αS-CM or oligomeric αSyn was significantly reduced by treatment with the LRRK2 inhibitor (MLi-2). Treatment with the TLR2 antagonist, CU-CPT22, reaffirmed that the neuroinflammatory responses of C20 via TLR2-specific αSyn oligomer or neuron-released αSyn in αS-CM would be mediated by TLR2 stimulation (Figure 4G,H, Appendix A). On the basis of the results we obtained using several cells, we concluded that LRRK2 kinase inhibitor could alleviate αSyn-mediated neuroinflammation pathology in PD.

### 3.3. Activation of Microglia by Dopaminergic Neuronal Cells Expressing αSyn Leads to Neurodegeneration Due to the Neuroinflammatory Response

The most severe result of neuroinflammation in patients with PD is the death of dopaminergic neuron [20]. Hence, we tried to observe the degeneration of dopaminergic neurons due to neuroinflammation caused by αSyn release from the dopaminergic neuron itself. We co-cultured C20 cells with dSH expressing Flag-αSyn or vector plasmid. We found that the co-culture of C20 with dSH transfected with Flag-αSyn showed a significantly lower neurite length in the latter cells (Figure 5A,B) than in the control cells, i.e., C20 co-cultured with dSH transfected with vector plasmid. 

We also confirmed that the cytotoxicity (Figure 5C), human TNFα release (Figure 5D), and NO levels (Figure 5E) were higher under the experimental conditions than under the control conditions. Interestingly, LRRK2-inhibitor (MLi-2) treatment of the co-culture of C20 with dSH transfected with Flag-αSyn ameliorated these aggravations (Figure 5). To reaffirm dopaminergic-neuronal death due to αSyn-mediated neuroinflammation, we examined the effect of conditioned media from C20 cells stimulated by P3C4 or αS-CM treatment on dSH and investigated whether the LRRK2 inhibitor could mitigate the death of dSH cells caused by C20 conditioned media. We observed that the media conditioned for P3C4 or αS-CM-activated C20 microglia cells increased the cytotoxicity and decreased the viability of dSH, and co-treatment with the LRRK2 inhibitor prevented neuronal toxicity and death by the reactive C20 microglia (Appendix A). On the basis of these results, we assumed that the pathological αSyn derived from dopaminergic neurons and neuroinflammation due to pathological αSyn would play an important role in the vicious cycle of PD progression.

### 3.4. LRRK2 Inhibitor Ameliorated the Neuroinflammation and Degeneration of Dopaminergic Neurons Triggered by Neuron-Released αSyn In Vivo

We attempted to combine and confirm all of the previously described results by conducting an experiment in which αS-CM and an LRRK2 inhibitor (MLi-2) were injected into the SN and peritoneum of mice, respectively; the experimental method proceeded as shown in Figure 6A. 

We observed that Vt-CM injection did not change the mobility of mice throughout the experiments, and αS-CM injection with vehicle administration significantly decreased the mobility of mice. However, administration of the LRRK2 inhibitor after αS-CM injection no longer worsened the mobility of mice (Figure 6B,C). In addition, the significant increase in the number of reactive microglia in the SN following the injection of αS-CM was moderately mitigated by the administration of LRRK2 kinase inhibitor (Figure 6D). Simultaneously, αS-CM injection with the administration of the vehicle elevated the optical densities of oligomeric αSyn and TLR2, phosphorylation of LRRK2, and TH in the SN (Figure 7).

In addition, the LRRK2 inhibitor alleviated the degeneration of dopaminergic neurons in the SN via the downregulation of microglial LRRK2 activation by TLR2-specific αSyn oligomers (Figure 7). In conclusion, neuron-released oligomeric αSyn-stimulated TLR2 may be responsible for PD pathogenesis, and LRRK2 inhibitors may be used as therapeutic drugs against neuroinflammation due to pathological αSyn propagation, as well as LRRK2-mediated pathology.

### 3.5. Association between the Upregulation of LRRK2 Activity and Increases in TLR2 and αSyn Oligomer Levels in the Human SN

To confirm our findings, we analyzed the lysates of human SN using immunoblotting and ELISA. There was no significant difference between the levels of the two αSyn oligomers (Figure 8A,B), but the phosphorylation of LRRK2 (pS1292) was significantly higher in the SN of patients with iPD (iPD SN) than in that of healthy controls (Ctrl SN) (Figure 8B,C).

Despite the similarity in the levels of TLR2, which were normalized to β-actin (Figure 8B,C), the TNFα levels in the iPD SN were higher than those in the Ctrl SN (Figure 8D). When these variables were compared at the individual level (Figure 8E), we found that higher levels of ASyO5-positive αSyn oligomers were related to higher pS1292, TLR2, and TNFα levels in iPD 2 and iPD 3. However, pS1292 in iPD 1 was not related to the elevation of TLR2 or TNFα levels, and the increase in filament αSyn oligomer was not responsible for TLR2 induction or LRRK2 kinase activity in Ctrl 2. Unfortunately, we could not observe TLR4 expression in either the Ctrl SN or iPD SN (Figure 8B). Further studies on SN lysates from patients is required owing to an insufficiency of samples in the current study. Nevertheless, we concluded that TLR2-specific αSyn oligomers upregulated LRRK2 kinase activity and elevated TNFα levels.

## 4. Discussion

Both LRRK2 and αSyn are associated with the etiology of PD, and previous studies reported that the clearance of αSyn in microglia is accelerated by the inhibition of LRRK2 activity [21]. Moreover, when using G2019S LRRK2 mutants, which show upregulated kinase activity, some studies observed an increase in the progression of αSyn pathology [22,23]. Considering the vicious cycle via αSyn-mediated neuroinflammation, LRRK2 activity acts as an important mediator of PD pathogenesis through pathological αSyn and its propagation. Although we demonstrated the crosstalk between neuron-derived pathological αSyn and neuroinflammation in microglia, we could not disregard the cell-to-cell transmission of αSyn between neurons and astrocytes, astrocytes and microglia, and other cell types in the brain [24,25,26]. In addition, TLR2 is not the only receptor of neuron-released oligomeric αSyn; TLR4 is also an important receptor for the clearance of neuron-mediated aggregates [27]. Recently, TLR2-specific αSyn oligomers have been reported to activate neuroinflammatory responses via the phosphorylation of NFATc2 by LRRK2 [28]. In this study, we revealed that TLR2-specific αSyn released from dopaminergic neuronal cells stimulates microglia via LRRK2 activation (Figure 2, Figure 3 and Figure 4) and that dopaminergic neuronal death was accelerated by neuroinflammation due to TLR2-specific αSyn released from dopaminergic neurons (Figure 5). LRRK inhibitors could ameliorate the neuroinflammation mediated by TLR2-specific αSyn oligomer and degeneration of dopaminergic neurons due to αSyn-mediated neuroinflammation in the mouse SN (Figure 6 and Figure 7). Moreover, two patients with iPD showed a positive association between TLR2, LRRK2 activity, TNFα, and αSyn oligomer (Figure 8). Therefore, these findings could be helpful in the development of LRRK2 kinase inhibitors for PD therapy.

Moreover, αSyn-targeting immunotherapy using specific antibodies as a PD drug has been reported by numerous researchers and investigated by the pharmaceutical industry [29,30,31]. The degradation and clearance of pathological αSyn using antibodies can decrease the levels of pathological αSyn in the brain, degeneration of dopaminergic neurons, and eventually, will prevent severe PD. Several reports [27,28] and this study suggest that LRRK2 inhibition may play a role in the protection of α-synucleinopathy. Hence, the co-administration of LRRK2 kinase inhibitors and pathological αSyn immunotherapy might exhibit synergistic and therapeutic efficiency by mitigating αSyn-mediated neuroinflammation.

## 5. Conclusions

We suggest that our findings would not only support the effect of LRRK2 kinase inhibitors on dopaminergic neuronal loss due to αSyn-mediated neuroinflammation, but LRRK2-inhibitor administration might also provide a synergistic therapeutic effect against PD with αSyn propagation.

## Figures and Tables

**Figure 1 cells-11-00861-f001:**
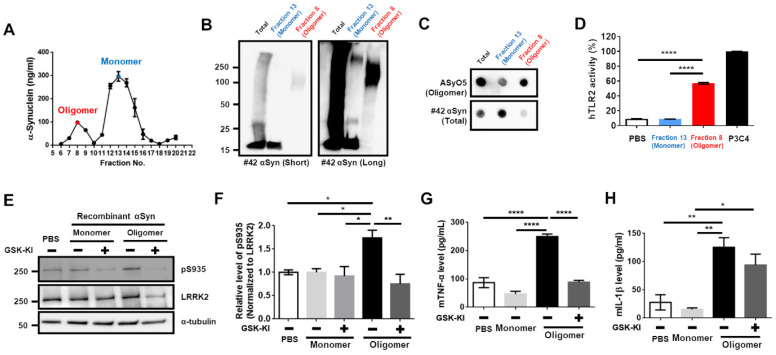
LRRK2 inhibitor mitigated LRRK2 activity and reduced mTNFα level that was upregulated by the TLR2-specific αSyn oligomer. (**A**) After separation of incubated αSyn pool using FPLC, the concentration of αSyn was assessed in each gel filtration fractions using sandwich ELISA. The oligomer (number 8, indicated with red-letter) and monomer (number 13, indicated with the blue letter) fractions were collected. Oligomer and monomer fractions and pre-injection mixture (Total) were subjected to Western blot (**B**) and dot-blot (**C**) analyses. (**D**) The human Toll-like receptor2 (hTLR2) activity induced by oligomer and monomer fractions was tested. PBS was used as the negative control, and P3C4 was considered as positive control. (**E**,**F**) After treating the cells with monomeric or oligomeric αSyn with or without the treatment with 1 μg/mL GSK-KI (LRRK2 kinase inhibitor, GSK2572815A) for 18 h, the cell lysates were analyzed using Western blotting. The levels of mTNFα (**G**) and mouse interleukin-1β (mIL-1β) (**H**) released were measured from (**E**). *n* = 4. * *p <* 0.05, ** *p <* 0.01, and **** *p* < 0.0001. All data are represented as the mean ± SEM.

**Figure 2 cells-11-00861-f002:**
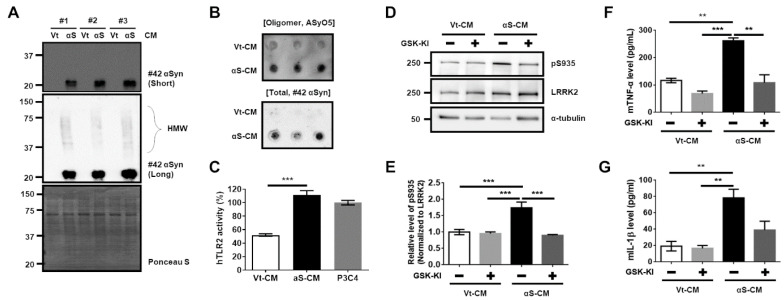
LRRK2 inhibitor hinders the LRRK2 activity and mTNFα release via the release of TLR2-specific αSyn from differentiated SH-SY5Y (dSH). The analyses of conditioned media of vector -transfected dSH (Vt-CM) and Flag-αSyn-transfected (αS-CM) using Western blotting (**A**) or dot-blot analysis (**B**) revealed an oligomeric formation of αSyn in αS-CM. The high molecular weight Flag-αSyn oligomer (HMW) band is indicated by the open curly bracket. # indicates a number of *n*. (**C**) The hTLR2 activity was higher in the treated αS-CM than in the treated Vt-CM. (**D**,**E**) The effect of the treatment of Vt-CM and αS-CM with or without GSK-KI for 18 h on BV2 was analyzed using Western blotting. The released levels of mTNFα (**F**) and mIL-1β (**G**) in the media were measured from (**D**). *n* = 3. ** *p <* 0.01 and *** *p <* 0.001. All data are represented as the mean ± SEM.

**Figure 3 cells-11-00861-f003:**
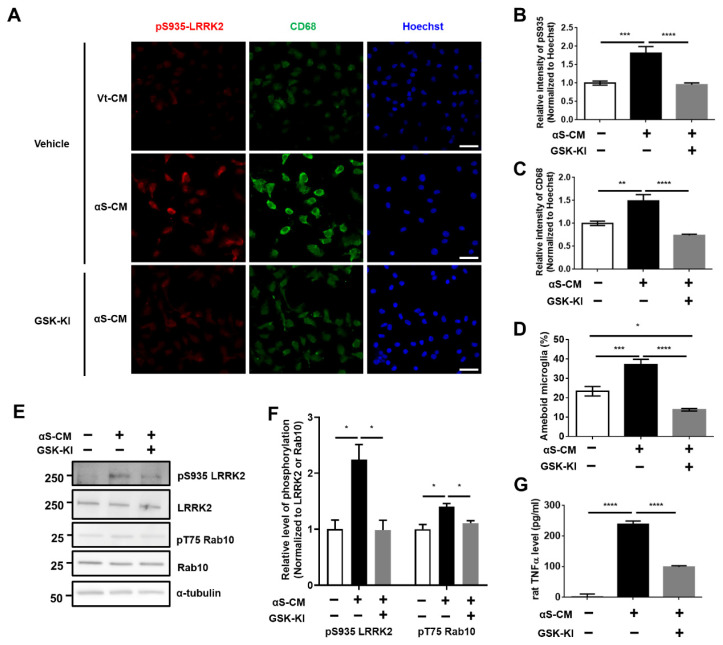
LRRK2 inhibitor ameliorated αSyn-induced neuroinflammation in rat primary microglia. (**A**–**C**) Immunostaining of pS935 and CD68, as markers of microglial activation, was assessed using their fluorescence intensities, and the microglia with ameboid morphology were counted (**D**). *n* = 3, approximately 30–60 cells for IF. (**E**,**F**) The change in LRRK2 activity induced by the treatment of αS-CM with or without GSK-KI was analyzed using Western blotting, and the released levels of rat TNFα from (**E**) were determined using ELISA (**G**). *n* = 3. * *p <* 0.05, ** *p <* 0.01, *** *p <* 0.001, and **** *p <* 0.0001. All data are represented as the mean ± SEM.

**Figure 4 cells-11-00861-f004:**
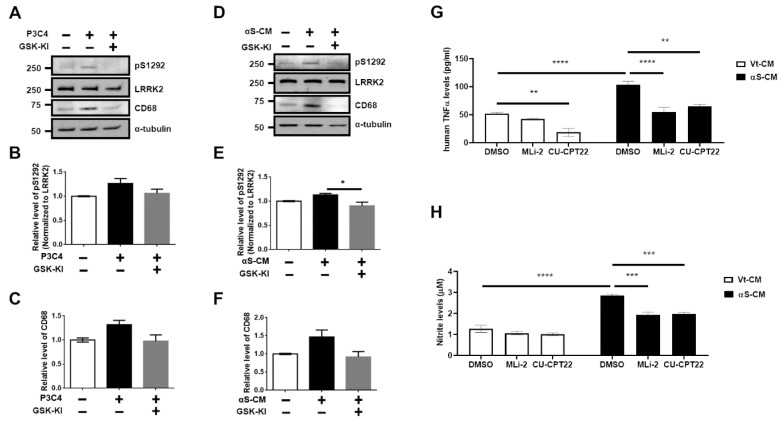
LRRK2 inhibitor decreased TLR2-mediated neuroinflammatory responses in human microglial cells. Western blot analysis of P3C4 treated with or without GSK-KI (**A**) helped assess the phosphorylation levels at S1292 in LRRK2 (**B**) and CD68 (**C**). Cell lysates obtained from the treatment of αS-CM with or without GSK-KI were subjected to Western blot analysis (**D**), and the levels of S1292 phosphorylation in LRRK2 (**E**) and CD68 (**F**) were estimated from the densitometric analysis of the immunoblots. *n* = 3. The released levels of human TNFα (**G**) and nitrite (**H**) were assessed from the culture media of C20 incubated with Vt-CM or αS-CM that was treated with DMSO (vehicle), MLi-2 (0.5 μM), or CU-CPT22 (10 μM) for 18 h. *n* = 4. * *p <* 0.05, ** *p <* 0.01, *** *p <* 0.001, and **** *p <* 0.0001. All data are represented as the mean ± SEM.

**Figure 5 cells-11-00861-f005:**
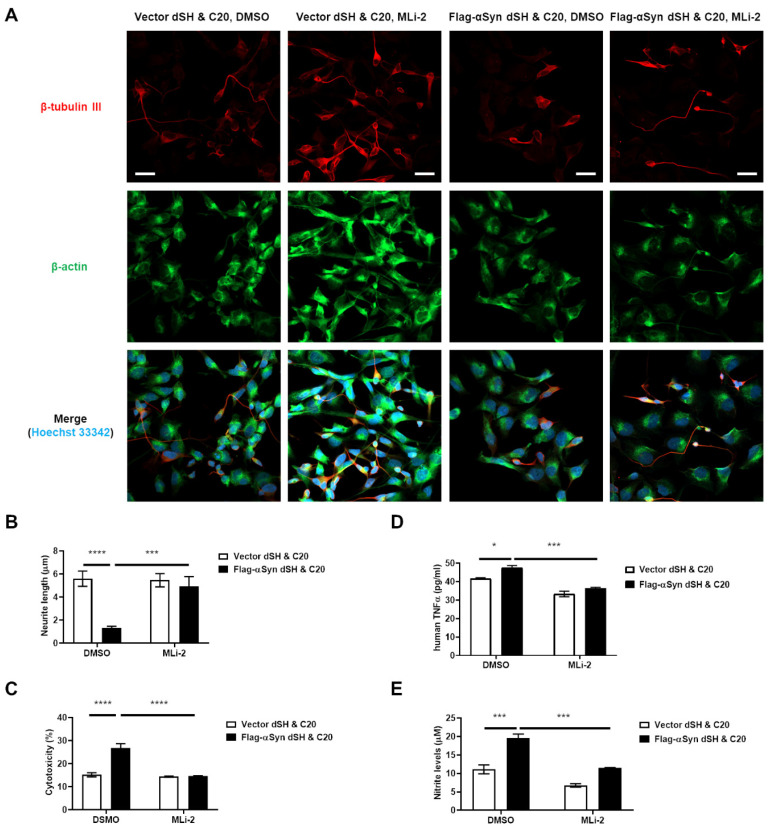
LRRK2 inhibitor alleviated the dopaminergic neuronal degeneration caused by neuroinflammatory responses, which are mediated by the dopaminergic neuron-released αSyn. (**A**,**B**) The neurite length of dSH was measured using the NeuronJ application in ImageJ program. Green: β-actin, Red: β-tubulin III, Blue: Hoechst 33342; approximately 21–23 cells for IF, *n* = 3. The measurement of LDH activity (**C**), hTNFα (**D**), and nitrite (**E**) in the culture media of (**A**) were analyzed. *n* = 3. * *p <* 0.05, *** *p <* 0.001, and **** *p <* 0.0001. All data are represented as the mean ± SEM.

**Figure 6 cells-11-00861-f006:**
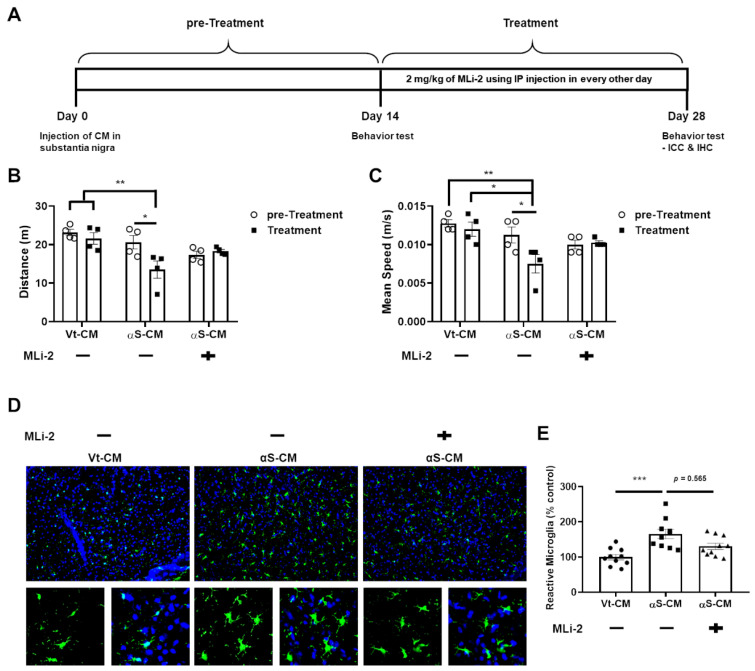
LRRK2 inhibitor prevented further progression of PD caused by neuron-released αSyn. (**A**) We schematized the plan of injection of VtCM or αS-CM and the administration of the LRRK2 inhibitor, MLi-2. We recorded the mobility of mice in terms of distance (**B**) and mean speed (**C**) on days 14 (pre-Treatment, circle) and 28 (Treatment, square), respectively. (**D**,**E**) The number of microglia that exhibited reactive morphology were analyzed. *n* = 4, approximately 10 sections for ICC. * *p <* 0.05, ** *p <* 0.01, and *** *p <* 0.001. All data are represented as the mean ± SEM.

**Figure 7 cells-11-00861-f007:**
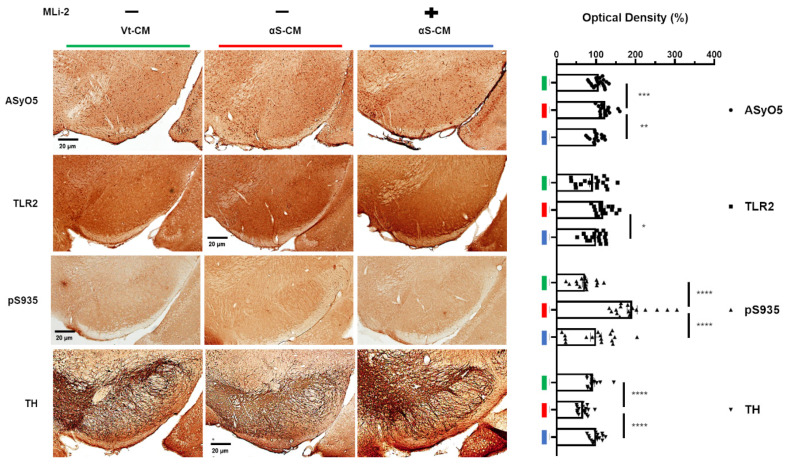
Administration of an LRRK2 inhibitor upregulated microglial activation by TLR2-specific αSyn oligomer. The optical density of αSyn oligomer recognized on the basis of AsyO5 antibody binding (circle), TLR2 (square), phosphorylation at S935 of LRRK2 (triangle), and the TH (reversed triangle) in the SN were estimated in the mice on day 28. *n* = 4, approximately 16–20 sections for IHC. * *p <* 0.05, ** *p <* 0.01, *** *p <* 0.001, and **** *p <* 0.0001. All data are represented as the mean ± SEM.

**Figure 8 cells-11-00861-f008:**
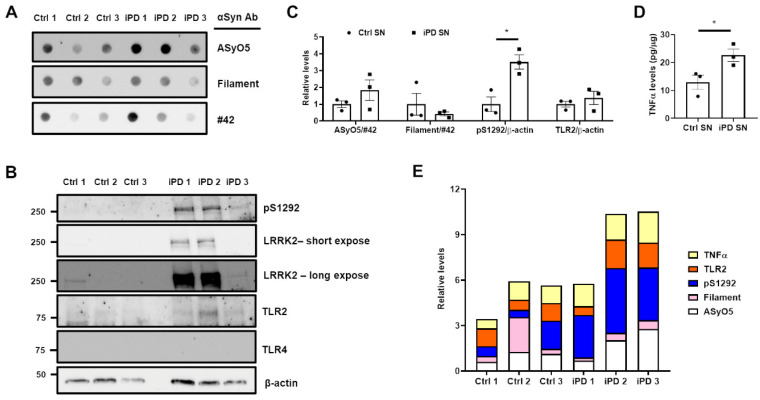
Post-mortem analyses of substantia nigra lysates form human patients. The SN lysates were subjected to (**A**) dot blot (50 μg of each lysate) and (**B**) Western blot (20 μg of each lysate) analyses, and membranes were analyzed using indicated antibodies. (**C**) The relative levels of oligomeric αSyn (normalized to #42 αSyn) and pS1292 or TLR2 (normalized to β-actin) in the SNs of control patients (Ctrl SN, circle) and idiopathic PD patients (iPD SN, square) were analyzed. (**D**) TNFα levels in SN lysate were measured using a human TNFα ELISA kit. (**E**) All relative data were aligned in a personal number. The colors indicate the following: yellow, TNFα; orange, TLR2; blue, pS1292; pink, Filament αSyn oligomer; white, AsyO5 αSyn oligomer. * *p <* 0.05. All data are represented as the mean ± SEM.

## Data Availability

The datasets generated and/or analyzed in this study are available from the corresponding author upon reasonable request.

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
