# Peer review of "LRRK2 Inhibition Mitigates the Neuroinflammation Caused by TLR2-Specific α-Synuclein and Alleviates Neuroinflammation-Derived Dopaminergic Neuronal Loss"

_cells, 2022, doi:10.3390/cells11050861_

Round 1

Reviewer 1 Report

This paper aims to prove that the inhibition of LRRK2 activity could alleviate the degeneration of dopaminergic neurons by mitigating excessive neuroinflammatory responses in microglia.

The manuscript is written comprehensively enough to be understandable despite of the complexity of the subject.

The paper stated the purpose, discussion and global implication are clearly stated and consistent with the rest of the manuscript; authors provided enough information in their discussion by using a good number of important articles talked about the subject.

The authors clearly described how the stimulator of neuroinflammation in patients with PD remains elusive and compared this fact with the previous studies which demonstrated that the release of α-synuclein (αSyn) from neurons is responsible for the activation of microglia via the toll-like receptors (TLRs), CD36, or P2X7 receptor. Then, they showed that previous studies revealed that a neuron-derived αSyn oligomer triggered a TLR2 specific activation of microglia and their neuroinflammatory response

The authors addressed their hypothesis and opinion in a reproducible way and proved their results through all the required experiments and analysis.

This study might be helpful for a direct application of LRRK2 inhibitor administration which might also provide an expectation of a synergistic therapeutic effect against PD with αSyn propagation.

It is recommended to use these papers to enrich the introduction and discussion:

Triggering of Inflammasome by Aggregated α–Synuclein, an Inflammatory Response in Synucleinopathies

  • https://doi.org/10.1371/journal.pone.0055375

Neuron-released oligomeric α-synuclein is an endogenous agonist of TLR2 for paracrine activation of microglia

  • DOI: 10.1038/ncomms2534

Leucine-rich repeat kinase 2 (LRRK2): a key player in the pathogenesis of Parkinson's disease

  • DOI: 10.1002/jnr.21949

No plagiarism has been detected.

Writing errors:

Row 46, 59: via (should be written in italic)

In vivo, in vitro (italic)

References: The authors followed perfectly the journal guidelines.

Author Response

We thank you for your pointing.

The first and third paper that you recommended were add in introduction and discussion. The second paper was already cited in this manuscript as a reference #14, because I am the one of author who particiapate to the second paper. 

Row 51 (third paper), 408 (first paper)

We corrected via in italic as your comment.

We edited 'In vivo' in reference #3 to italic 'In vivo', and rest of in vivo or in vitro was written to notice manufacturers, InvivoGen and Invitrogen.

And we requested the language editing to Editage before the submission of this manuscript, twice.

Hence, we attached the certification of language editing from Editage.

Reviewer 2 Report

The report by Ho et al. examined whether LRRK2 inhibition reduced microglial responses to neuroinflammation, caused by neuron-released αSyn. Authors investigated responses provoked by toll-like receptor 2 (TLR2)-positive αSyn of neuronal cells using an LRRK2 inhibitor in mouse glioma cells, rat primary microglia, and human microglia cell line and the effects of LRRK2 inhibitor in the co-culture of ectopic αSyn-expressing human neuroblastoma cells and human microglia cells and in mouse models by injecting αSyn. They also tested the association between LRRK2 activity and αSyn oligomer and TLR2 levels in the substantia nigra tissues of patients with idiopathic PD. TLR2-specific αSyn increased LRRK2 activity and neuroinflammation, and the LRRK2 inhibitor ameliorated neuroinflammatory responses in microglia cells, alleviated neuronal degeneration along with neuroinflammation in the co-culture, and blocked the further progression of locomotor failure and dopaminergic neuronal degeneration caused by TLR2-specific αSyn in mice. LRRK2 phosphorylation was increased in PD patients showing αSyn-specific high TLR2 level. Their results suggest the application of LRRK2 inhibitors as a novel therapeutic approach against αSyn-mediated PD progression.

Experimental design is sound, introduction is informative. Metyhods and results are correctly displayed and discussion is balanced. references are appropriate

Author Response

We appreciate for your detailed reviewing and comments.

And we already requested and served the language editing by Editage before this submission, twice.

Hance, we attached the certification of language editing service from Editage.
